# Content aware image restoration improves spatiotemporal resolution in luminescence imaging

Tobias Boothe [1✉], Mario Ivanković[1], Markus A. Grohme[2], M. Andrea Markus[3], Christian Dullin[3,4,5], Xingbo Xu[6] & Jochen C. Rink [1✉]

Luminescent reporters are due to their intrinsically high signal-to-noise ratio a powerful labelling tool for microscopy and macroscopic in vivo imaging in biomedical research. However, luminescence signal detection requires longer exposure times than fluorescence imaging and is consequently less suited for applications requiring high temporal resolution or throughput. Here we demonstrate that content aware image restoration can drastically reduce the exposure time requirements in luminescence imaging, thus overcoming one of the major limitations of the technique.

[1] Max Planck Institute for Multidisciplinary Sciences, Am Fassberg 11, 37077 Göttingen, Germany. [2] Max Planck Institute for Molecular Cell Biology and Genetics, Pfotenhauerstrasse 108, 01307 Dresden, Germany. [3] Max Planck Institute for Multidisciplinary Sciences (City Campus), Translational Molecular Imaging, Hermann-Rein-Str. 3, 37075 Göttingen, Germany. [4] Department for Diagnostic and Interventional Radiology, University Medical Center of Göttingen, Robert-Koch-Straße 40, 37075 Göttingen, Germany. [5] University of Heidelberg, Translational Lung Research Center (TLRC), German Center for Lung Research (DZL), Diagnostic and Interventional Radiology, Im Neuenheimer Feld 130.3, 69120 Heidelberg, Germany. [6] Department of Cardiology and Pneumology, University Medical Center of Göttingen, Robert-Koch-Straße 42a, 37075 Göttingen, Germany. ✉email: tobias.boothe@mpinat.mpg.de; jochen.rink@mpinat.mpg.de

In biomedical research, fluorescence microscopy is widely used for the specific visualisation of proteins, organelles, cells, organs or entire organisms[1]. Despite its versatility and applicability on many scales, this imaging technique exposes the sample to potentially harmful excitation light and is also sensitive to auto-fluorescence artefacts[1]. In contrast, bioluminescence imaging exploits the light emitted by a chemical reaction between a luciferase enzyme and its substrate (luciferin). Luciferase enzymes originate from approximately 10,000 bioluminescent species across the tree of life. Well characterised natural or various biotechnologically-optimised luciferase reporters are available[2]. Luminescence micro-scopy does not require excitation light and is highly specific due to the practical absence of spontaneous photon emission in biological samples. The technique is therefore especially powerful for imaging photosensitive or highly autofluorescent samples[3,4]. However, a major drawback of bioluminescence imaging are the low signal intensities, which typically require much longer exposure times in comparison with fluorescence imaging. This practically restricts luminescence microscopy applications to immobile samples and imposes throughput limits on high content screening applications. Although shorter exposure times could, in principle, remedy these shortcomings, the inevitable decrease in the signal-to-noise ratio limits the practical utility of this approach. Recently, Weigert et al. presented an approach for denoising fluorescent microscopy data by utilising deep neural networks, through which signal-to-noise ratios could be enhanced post acquisition[5]. In this method, we train a deep neural network with image pairs consisting of low and high signal-to-noise recordings. From this, the network learns to denoise images of low signal quality and ultimately enhances the image contrast in a content aware manner. Here we show that this content aware image restoration (CARE) can similarly restore luminescence recordings without compromising image quality, allowing exposure time reductions up to 1000-fold. By overcoming one of the major limitations of luminescence imaging, our results significantly expand the practical utility of luminescence imaging.

## Results

Convolutional networks have demonstrated strong denoising capabilities in fluorescence microscopy[5,6]. To explore their corre-sponding utility in luminescence imaging, we trained a CARE network with luminescence recordings of different exposure times. In our experimental setup, human tissue culture cells expressing untargeted NanoLuc (Nluc) luciferase required 60 s exposure time on a commercially available luminescence imaging system to achieve satisfactory signal-to-noise ratios. These recordings served as ground truth for restoring signals from exposure times as short as 0.5 s. Training networks on short and long exposure image pairs denoised and restored short exposure images to virtually ground truth quality (Fig. 1a and Supplementary Fig. 1). To quantify the quality of the restorations, all images were automatically thre-sholded resulting in binary masks representing the objects of interest. The disagreement of masks obtained from input, restoration and ground truth images was quantified with the mean absolute error (MAE). The restorations from the aforementioned recordings display a MAE that is similar to the noise between object masks from two subsequently acquired images at 60 s exposure time ("technical noise").

Another commonly used strategy for decreasing exposure times is pixel binning, which sacrifices resolution for a higher signal-to-noise ratio. CARE has previously shown strong capabilities in restoring undersampled z-resolution in 3D recordings[5]. We sought to transfer this 1D Z-resampling power to the 2D XY-dimension by training a CARE network with binned recordings ($2 \times 2$, $4 \times 4$, $8 \times 8$) at low exposure times and unbinned long exposure time ground truth images. We demonstrate that the resolution lost by

pixel binning can be restored even from $8 \times 8$ binned recordings (Fig. 1b and Supplementary Fig. 2). Therefore, pixel binning and subsequent image restoration provide a further layer of reducing luminescence exposures, with the additional benefit of enhanced visibility of the residual signal in the raw images at short exposures.

Thus far, we described approaches that require long exposure time recording for generating ground truth training data. Technical or biological constraints, however, can make long exposures problematic. Therefore, we additionally explored the capabilities of noise2noise image restoration in which only noisy image pairs are used as training data[7,8]. From such training pairs the CARE network is able to identify and remove statistical noise without the need for long exposure images as ground truth. In our setup exposure times as low as 5 s per image were sufficient to create training data that allowed signal restoration to a degree comparable to that of 60 s exposure times (Fig. 1c and Supple-mentary Fig. 3).

One of the major biomedical research applications of lumines-cence imaging at present is the non-invasive imaging of rodent disease models[9]. Despite its popularity, luminescence in vivo imaging on the macroscopic scale suffers from the same technical constraints as luminescence microscopy. We therefore turned to in vivo imaging of a conventional firefly luciferase reporter in live mice to evaluate our method. We demonstrate that CARE networks trained with short and long exposure image pairs (noise2signal) or short exposure image pairs only (noise2noise) both achieved signal restoration of noisy short exposure data comparable to long exposure ground truth images (Fig. 1d and Supplementary Fig. 4). In our system, we could achieve a shortening of exposure times of up to 10-fold without major compromises on segmentation quality. Further, these results demonstrate that CARE can also be applied to 2-dimensional (single-plane) image data as it is typically obtained from non-invasive in vivo imaging systems.

Overall, our findings underline the suitability of our approach for various sample types across biological scales and demonstrate a substantial shortening of luminescence exposure times that are achievable with CARE.

Image restorations performed with CARE - just like any machine learning algorithms - perform best when the training data accurately represents the data to be restored[5]. In practise, this often entails frequent retraining of the models whenever experimental conditions change and the recording of training data can thus quickly become a bottle-neck. We therefore eval-uated the co-transfection of fluorescently labelled proteins requiring comparatively short exposure times as ground truth for their luminescent labelled equivalent. To test this training approach we transiently co-transfected cells with NanoLuc tagged Histone 2B (H2B-Nluc) and eGFP-fused H2B plasmid DNA. We continued to compare the restoration quality of the CARE net-works when training with image pairs of short exposure Nluc-H2B signal and eGFP-H2B signal or long exposure Nluc-H2B signal as ground truth respectively. When analysing the by-pixel signal correlation between objects segmented from restorations and ground truth signal, we show that restoration from lumi-nescence signals based on training to fluorescent ground truth is virtually indistinguishable to restorations obtained from training to long exposure luminescence ground truth signals (Fig. 2a and Supplementary Fig. 5). Note that luminescence exposure times as short as fluorescent ground truth exposure times provide suffi-cient signal from the tagged H2B proteins for an accurate restoration, thus making high-throughput applications of lumi-nescence imaging feasible.

Due to the rather low signal emission, luminescence microscopy has been predominantly applied to whole tissue or organism imaging since the signals are often too weak for the live imaging of subcellular dynamics. To assess restoration qualities in this context,

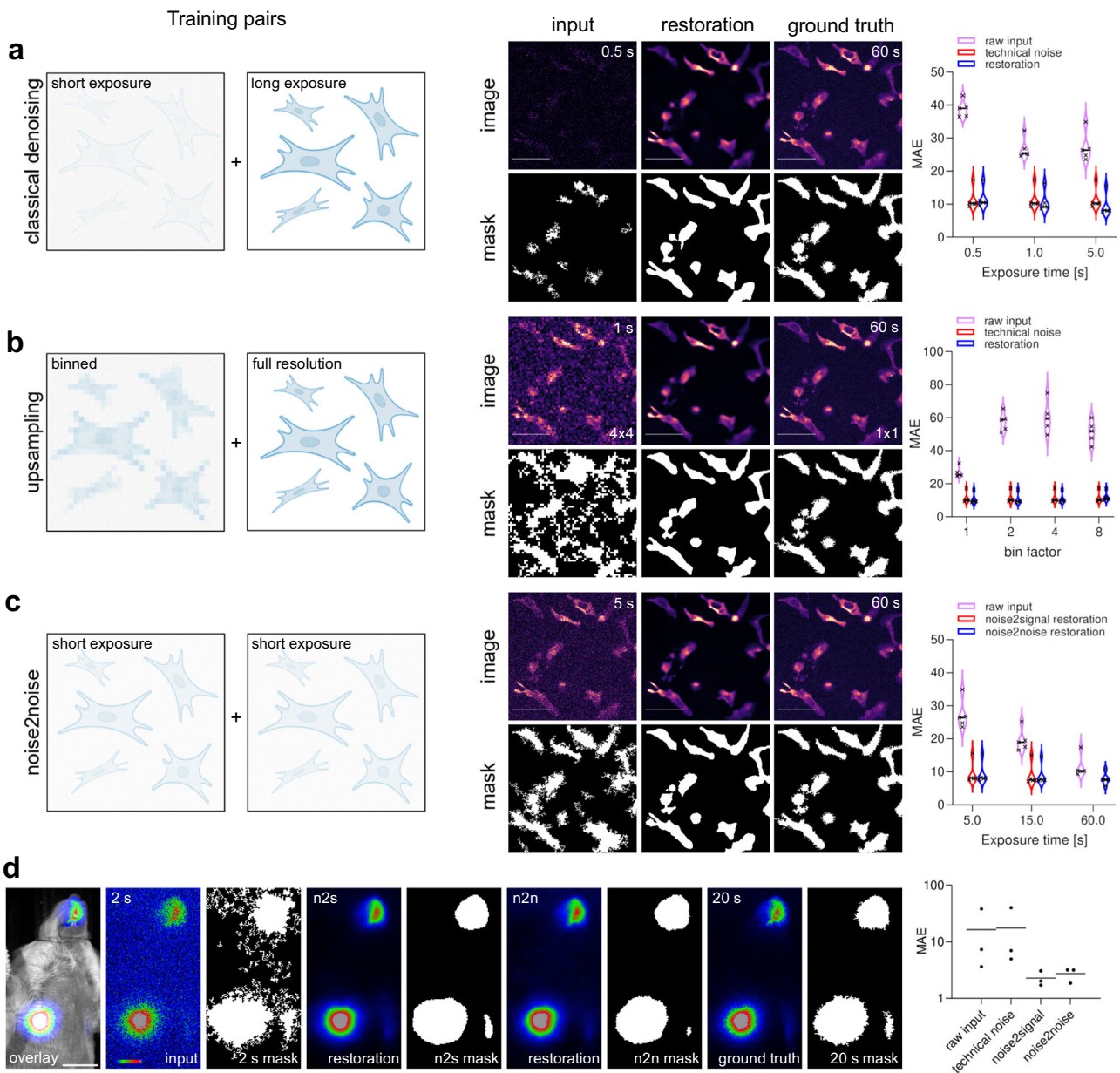

**Fig. 1 Luminescence image restoration. a** Content aware image restoration (CARE) allows the restoration of images obtained with short exposure times to images with a signal-to-noise ratio that is normally achieved with much longer exposure times. **b** CARE can be used to upsample and restore binned images to full resolution. **c** CARE can be utilised for noise2noise image restorations by training a network with image pairs of short exposure times only. **a–c** Schematics for CARE training pairs are illustrated. Restorations were performed on previously unseen data. Binary images represent automatically segmented objects from the respective micrograph. The mean absolute error (MAE) between object masks was used as a quantitative readout to compare restorations with ground truth (lower is better). Scale bars = 10 μm, horizontal bars indicate median values (**d**) Firefly luciferase expressed in a mouse model via AAV delivery. 2 s exposure (input) leads to poor segmentation compared to 20 s exposure (ground truth) as assessed by MAE between masks. CARE can reliably restore 2 s exposure images via noise2signal or noise2noise trained networks to ground truth level. Scale bar = 1 cm, N = 3 animals (**a–d**) Violin plots depict data distribution, horizontal bars show median values. N = 5 ROIs.

we fused NanoLuc to the outer mitochondrial membrane protein TOMM20[10] (TOMM20-Nluc) and used mitochondrial dynamics as model system[10]. Due to the continuous remodelling of the mitochondrial network, the recording of long exposure ground truth data is practically impossible. As previously shown (Fig. 2a), fluorescently labelled proteins can act as ground truth while keeping ground truth recordings at short exposure times to avoid motion artefacts. Analogous to the H2B-Nluc/eGFP co-transfection (see above), we trained a network on recordings of cells co-transfected with TOMM20-Nluc and TOMM20-NeonGreen. In this approach we demonstrate that luminescence signals

obtained from exposure times as short as 2 s can be reliably restored to the fluorescently labelled TOMM20-NeonGreen ground truth equivalent (Fig. 2b). Mitochondria dynamics, which include fusion and fission events, can be a response to cellular stress and are therefore an important indicator of cellular health[11]. To test if the restoration quality is sufficient to characterise mitochondrial phenotypes, we exposed cells to Rotenone - a well characterised respiratory inhibitor that induces mitochondrial fragmentation[12]. We analysed mitochondrial morphology by quantifying the organelle's circularity and aspect ratio. Despite the challenges associated with segmenting diffraction limited structures

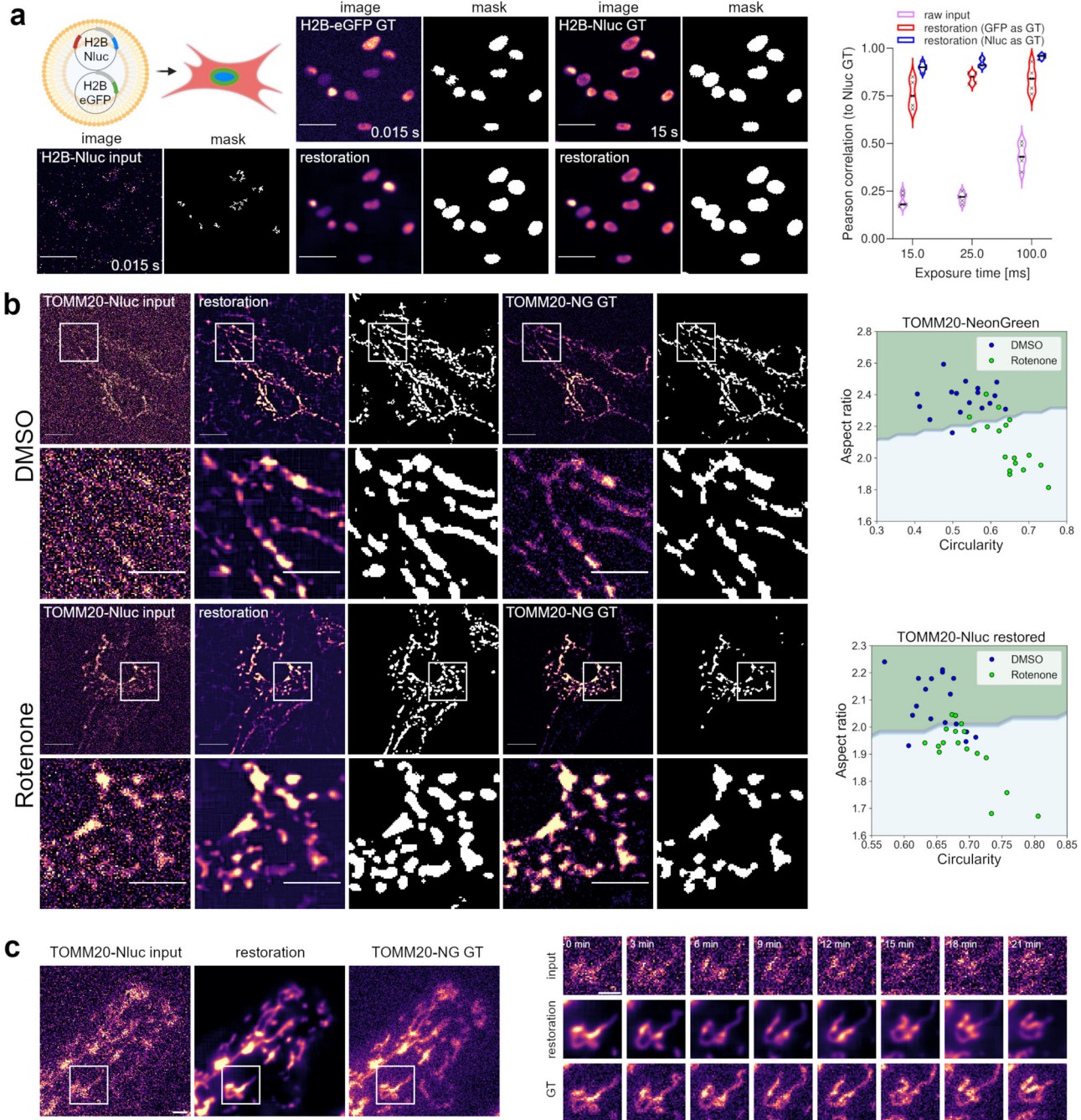

**Fig. 2 Restoration of luminescence signals to fluorescent ground truth. a** Fluorescently labelled proteins can act as ground truth to reduce exposure times required for training data acquisition. H2B-Nluc and H2B-eGFP were co-transfected and training pairs were generated by capturing short exposure luminescence and short exposure fluorescence signals. Long exposure luminescence signals were recorded for quality control. The Pearson correlation of pixel intensities under masked objects was used as a restoration quality readout. (No statistically significant difference was detected between the restoration modalities. $p = 0.209$, two-way ANOVA, $N = 5$ ROIs, individual measures, technical replicates) Scale bars = 10 μm. Violin plot depicts data distribution, horizontal bars show median values. **b** Restoring luminescence signals from diffraction limited structures. Mitochondria were labelled with TOMM20 fusion proteins and a network was trained with luminescence (2 s exposure time) and corresponding fluorescence image pairs. Restorations from luminescence signals allowed for a similar phenotype classification compared to classifications from fluorescently labelled mitochondria. Rotenone treatment was used to induce mitochondrial fragmentation for phenotype scoring. The decision boundary was fitted by linear regression. Scale bars = 5 μm (**c**) Restoration of mitochondria remodelling in time lapse recordings. Shown are single optical slices. Network training and restoration conditions equivalent to panel (**b**). Scale bars = 2 μm.

on a widefield microscope, we show that restorations of Rotenone treated cells can be distinguished by a fitted linear decision boundary from their DMSO treated controls. Separation of mito-chondrial morphology performed equally well on TOMM20-Nluc signal reconstructions as compared to reconstructions based on

fluorescent TOMM20-NeonGreen ground truth of the same cell (Fig. 2b). Additionally, we demonstrate that the highly dynamic remodelling behaviour of mitochondria at a steady state can be accurately restored from noisy luminescence time-lapse recordings (Fig. 2c and Supplementary Fig. 6).

Therefore, our results demonstrate that CARE enables luminescent imaging and analysis of dynamic intracellular compartments that were previously inaccessible due to long exposure time requirements.

## Discussion

The present study demonstrates the utility of CARE in luminescence imaging. Specifically, the previously introduced CARE method enabled the reliable restoration and denoising of luminescence micro- and macrographs with initially low contrast, thus shortening required exposure times up to 1000-fold under our experimental conditions.

We successfully applied the original CARE network in noise2noise restorations, which is useful when long-term exposures of ground truth are not possible due to biological or technical constraints. Especially in non-dedicated luminescence imaging setups, light pollution by external sources such as instrument LEDs or room light can impact long term exposures and thus greatly complicate the acquisition of the ground truth data.

We further showed that the CARE network architecture can be used to reliably upsample binned images thus lifting the constraint previously imposed on image resolution. Furthermore, upsampling is useful, when camera read out times are a constraining factor in highly dynamic events as the camera chip read out times for binned images are much shorter than those at full resolution. This is especially interesting for rapid imaging in fluorescence microscopy to which this procedure can be equally applied. It is important to stress that deep learning based restoration methods are - like any restoration method - prone to artefacts[13], which can be assessed computationally, as outlined in the original CARE publication[5].

We finally demonstrate that fluorescence signals can also act as ground truth for luminescence signals in particularly sparse and spatially distinct structures. This further shortens exposure times required for training data acquisition. It is especially useful in very challenging samples that require short exposure times already for training data generation because of biological or practical constraints. With CARE we demonstrate one neural network architecture that can be applied to restore signals in luminescence images. In principle, other networks dedicated to image restoration are potentially equally suitable[6,14].

Together, the application of CARE to luminescence imaging shortens recording times from hours to minutes or from minutes to seconds, which in turn entails a significant expansion of the possible applications of luminescence imaging. Our time lapse recordings of mitochondrial dynamics (Fig. 2c) provide a first proof of principle: Subcellular dynamics have so far been largely out of reach of luminescence reporters due to the associated frame rate constraints. Since luminescence reporters eliminate the phototoxicity of the excitation light in fluorescence imaging, our method may thus enable the long-term imaging of particularly sensitive tissue culture models or the imaging of subcellular dynamics in tissues with high autofluorescence.

In the case of in vivo luminescence imaging, the exposure time reductions are similarly compelling. The up to 10-fold exposure time reduction that we achieved on the basis of a standard firefly luciferase reporter and a conventional imaging system (Fig. 1d) can certainly be reduced further via the use of enhanced luciferase reporters and optimised imaging setups[15]. Hence we suggest, that exposure times in the sub-second range are feasible, which in turn might enable the luminescence imaging of free-ranging rodent models without the need for anaesthetics.

Overall, our method lifts the long exposure time requirements as a major current limitation of luminescence imaging, thus dramatically broadening the application range of luminescence imaging in biomedical research.

## Methods

**Plasmids and molecular cloning**. DNA constructs containing open reading frames (ORFs) encoding H2B-Nluc and TOMM20-Nluc C-terminal fusion proteins were commercially synthesised (Eurofins Genomics). The ORFs were cloned into pBI-CMV4 (Takara Bio) utilising 5'-NheI and 3'-SalI restriction sites with reagents from New England Biolabs following standard procedures yielding the final plasmids used for subsequent transfections.

Cytoplasmic NanoLuc was expressed by transfecting U2OS cells with pcDNA3.1-NL plasmid. Plasmid pcDNA3.1-NL was obtained from Addgene (Addgene plasmid #113442). Fluorescently labelled H2B was expressed by transfecting pEGFP-N1-H2B plasmid. H2B-GFP was obtained from Addgene (Addgene plasmid # 11680). Fluorescently labelled Tomm20 was expressed by transfecting pN1-TOMM20-mNG plasmid (Addgene plasmid # 129347).

**Cell culture**. All cell lines were cultured at 37 °C, 90% humidity and 5% $CO_2$. U2OS cells were cultured in DMEM medium (Gibco Cat# 31885-023) supplemented with 10% v/v FBS (Anprotec Cat# AC-SM-0033), 100 U/ml Penicillin-Streptomycin (Gibco Cat# 15140-122). Hela CCL-2 cell lines were cultured in DMEM media (Corning Cat# 15-013-CV) supplemented with 10% v/v FBS, 100 U/ml Penicillin-Streptomycin and 10 mM L-glutamine (Gibco Cat# 25030-024).

**Transfections**. All plasmid transfections were performed at 70% confluency using Lipofectamine 3000 transfection reagent using 2 µl Lipofectamine and 2 µl P3000 reagent/µg DNA (Thermo Fisher Scientific). For transfecting 30 mm dishes or 75 cm$^2$ flasks, a total of 2.5 µg or 25 µg of plasmid DNA were used, respectively. For double transfections, plasmids were combined equally while retaining absolute amounts used for single transfections.

**Fluorescence-activated cell sorting**. To ensure coexpression in experiments that were used to train luminescence signal on fluorescent ground truth signal, luminescence fusion proteins were cloned into a bidirectional vector (pBI-CMV4) that also expresses dsRed2 as an expression control for the luminescent fusion protein (see section 'plasmids and cloning' for details).

For experiments in which co-expression of fluorescently labelled H2B and Nluc conjugated H2B was required, a transfected, confluent 75 cm$^2$ flask of HelaCCL cells was harvested. Cells were resuspended in complete DMEM and kept on ice and FACsorted at 5 °C: eGFP$^+$/dsRed2$^+$ double positive populations were isolated by FACS using a SONY Cell Sorter SH800 with a 100 µM microfluidics sorting chip. Scatter characteristics utilising FSC-A/BSC-A were used to exclude debris and FSC-A/FSC-W to exclude doublets. eGFP$^+$/dsRed2$^+$ cells were sorted using the "Semi-Purity" sort mode with a sensor gain of 32% for dsRed2 (FL3-600/60) and eGFP (FL2-525/50). During the sort, 488 nm and 561 nm lasers were active. Per 35 mm glass bottom dish 300,000 double positive cells with comparatively high expression ratios of both eGFP and dsRed were seeded.

**Luminescence assays**. For cells expressing cytosolic NanoLuc or H2B-NanoLuc fusion proteins, Nano-Glo Endurazine (Promega) was used as a substrate at a 1x final concentration. Cells were imaged 1 h after addition of the substrate.

For cells expressing TOMM20-Nluc fusion protein, Nano-Glo Vivazine (Promega) was used as a substrate at a 1x final concentration. Cells were imaged 1.5 h after addition of the substrate.

For in vivo experiments, firefly luciferase (Fluc) expression was driven by a ubiquitous promoter in an AAV vector backbone. AAV was packaged and purified by following the previously published protocol[16]. Vivo Glo Luciferin (Promega) was intraperitoneally injected into each animal at a concentration of 150 mg/kg body weight.

All animal in-vivo procedures were performed in compliance with the guidelines of the European Directive (2010/63/EU) and the German ethical laws and were approved by the administration of Lower Saxony, Germany (# G18/2773). The mouse strain B6N-Tyrc-Brd/BrdCrCrl (Charles River) was used for this study. Sex and age were not considered in this study and is therefore not reported.

**Microscopy**. All images were recorded using Olympus' LV200 bioluminescence imaging platform. eGFP and NeonGreen were excited through a 470/11 nm bandpass filter. Emission for eGFP and NeonGreen was collected with a 525/25 nm bandpass filter. Luminescence was detected without any emission filter. An Olympus 20x NA 0.8 UPLXAPO objective was used to image cytoplasmic NanoLuc and H2B-Nluc/eGFP signals. An Olympus 100x NA 1.5 UPLAPO OHR objective was used to image TOMM20-Nluc/mNeonGreen. For signal detection, an Andor iXon 888 Ultra EM-CCD camera, deep cooled to −85 °C at a 1 MHz readout rate with an EM gain of 300 was used. Cells were incubated with a Tokai Hit stage top incubator providing full environmental control.

Cells for imaging experiments were cultured in 30 mm glass bottom dishes (ibidi, Cat# 81158). Cells were imaged in $CO_2$-independent, phenol-red free, L15 Leibovitz media (Thermo Fisher Cat# 21083027) supplemented with 10% v/v FBS

and 100 U/ml Penicillin-Streptomycin. L15 Leibovitz imaging media for Hela CCL cells was additionally supplemented with 10 mM L-Glutamine.

**In vivo imaging**. Bioluminescence images were acquired within the first 15 min following luciferin injection using the IVIS Spectrum In Vivo Imaging System (Perkin Elmer). The Andor iKon DZ436 CCD Camera was deep cooled to −90 °C. The bioluminescence modality was used with the field of view (FOV) set to B and the subject height set to 1.5 cm. Each FOV was acquired with exposure times as indicated.

**Data recording, deep neural network training and image restoration**. For training and restoration, the previously published CSBDeep package v0.6.0 was used (http://csbdeep.bioimagecomputing.com). A detailed documentation of this software is available at http://csbdeep.bioimagecomputing.com/doc.

In general, the presented method consists of 4 steps (Supplementary Fig. 7): First, training data pairs need to be recorded (Fig. 1a–c). They should be as similar as possible to the experimental data that will be restored later. Second, the CARE network needs to be trained. Third, the experimental data can be recorded, which consists only of noisy image data. Fourth, the experimental data can be restored using the previously trained CARE network.

We performed all training and prediction pipelines using our publicly available docker container that can be obtained at https://hub.docker.com/r/tboo/csbdeep_gpu_docker. Generalised Python scripts that can be used with this container for preparing training data, network training and prediction are available at https://gitlab.gwdg.de/rinklab_public/lumicare.

Networks were trained on a Lenovo ThinkSystem SR670 server equipped with two Intel Xeon Gold 6234 CPUs, 768GB RAM and four NVIDIA Tesla V100 32GB GPUs.

Networks were trained for each low-high signal condition separately, providing best restoration performance.

All tissue culture training data were recorded in 3D (XYZ) and all networks were trained as 3D networks. For acquisition of training data, input (low signal) condition(s) and ground truth/target condition (high signal) were imaged consecutively per plane before proceeding to the next z-plane. Training data for noise-to-noise training was obtained by taking 2 consecutive images with identical imaging parameters. To train networks for upsampling, training data was obtained by software binning using the "bin" function in Fiji. These binned images were subsequently upsampled without interpolation to match the pixel dimensions of the respective ground truth image. In vivo training data were recorded in 2D (XY) with the subsequent acquisition of two 2 s exposures and a 20 s ground truth exposure. To increase training data complexity each raw image was computationally rotated 3 times in 90° increments and subsequently each of the resulting stacks was computationally mirrored horizontally resulting in an 8-fold increase of available training data.

Supplementary Table 1 summarises the key parameters used for training data acquisition, training data preparation and network training. Networks were trained with probabilistic per pixel prediction (probabilistic=True). All other network parameters were used in default settings.

For all conditions, the trained models were used to restore images by tiling the respective stacks 2 × 2 × 2 in XYZ using the csbdeep API.

**Image analysis**. All image processing and analysis was performed using the ImageJ distribution Fiji v2.3.0[17]. To quantify the disagreement between ground truth images of cells expressing cytoplasmic NanoLuc and respective restoration results from low signal images, the mean absolute error (MAE) between cell masks was determined using the SNR plugin v06.05.2011[18]. To obtain the masks, all images were converted to 16 bit, thresholding was applied using the "Mean" auto thresholding function with enabling the "dark background" option. Resulting particles were filtered by size (>100 pixels). The resulting masks were inverted and used with the SNR plugin to quantify the MAE between the ground truth masks as the reference images and the corresponding restoration or input masks as test images.

For quantifying the disagreement between ground truth in vivo Fluc images and the respective restoration, all images were thresholded using the "Mean" auto thresholding function enabling the "dark background" option. Resulting particles were filtered by size (>10 pixels). The resulting masks were inverted and used with the SNR plugin to quantify the MAE between the ground truth masks as the reference images and the corresponding restoration or input masks as test images.

To quantify the disagreement between ground truth images of cells expressing labelled H2B and respective restoration results from low signal images, the Pearson correlation between intensities of masked nuclei was determined. For that purpose, ground truth images were thresholded using the "Triangle" auto thresholding function enabling the "dark background" option. Resulting particles were filtered by size (>5 pixels). The resulting masks were inverted and used as masks for determining the Pearson correlation of raw pixel values between an input/restoration image and the respective ground truth images utilising the "Coloc2" plugin v3.0.5 in Fiji.

For quantifying mitochondrial morphology all images (including restorations) were maximum projected along Z and a rolling ball background subtraction

(radius = 5 pixels) was applied using Fiji. To quantify the disagreement between ground truth images of cells expressing labelled Tomm20 and respective restoration results from low signal images, a pixel classifier to segment mitochondria was trained using ilastik software v1.3.3[19]. For training this pixel classifier, a subset of mitochondria and background was annotated in 10 images total (5 DMSO control, 5 Rotenone treated) of cells expressing TOMM20-NeonGreen (ground truth images). This trained classifier predicted accurate mitochondria masks in all images used for analysis. The resulting masks were subsequently analysed for their morphology with Fiji by setting an object size threshold (>20 pixels). Circularity and aspect ratio of these segments was measured via the 'shape' measurements module. The values displayed in Fig. 2 are object measurement mean values per image analysed.

**Statistics and reproducibility**. Statistical tests were conducted with GraphPad Prism software v 9.5.1. All results reported were reproducible in independent samples. For cell culture experiments different passages were considered as independent samples. For in vivo experiments, different animals were defined independent samples.

**Reporting summary**. Further information on research design is available in the Nature Portfolio Reporting Summary linked to this article.

## Data availability
Source data underlying figures are provided in Supplementary Data 1. All raw data are available upon request to the corresponding authors.

## Code availability
The source code used in this study is available at https://gitlab.gwdg.de/rinklab_public/lumicare.

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

## Acknowledgements
The authors would like to thank Britta Schroth-Diez and Jan Peychl from the Light Microscopy Facility at the Max Planck Institute for Molecular Cell Biology and Genetics for generous support and encouragement during the first proof of concept experiments for this study. We would further like to thank Dajana Burghardt for technical assistance with molecular cloning. pcDNA3.1 NL was a gift from Erich Wanker. pEGFP-N1-H2B was a gift from Geoff Wahl. pN1-TOMM20-mNG was a gift from Yasushi Okada. Mammalian cell lines were a gift from Stefan Jakobs. Schematics in Figs. 1a–c, 2a and S3 were created with BioRender.com.

## Author contributions
T.B. and M.A.G. developed the original concept of this work. T.B., M.I., M.A.G., M.A.M., C.D., X.X. and J.C.R. contributed to experimental design. T.B., M.I., M.A.M. and C.D. performed experiments. T.B. wrote the initial manuscript. T.B., M.I., M.A.G., M.A.M., C.D., X.X. and J.C.R. edited the manuscript.

## Funding

## Competing interests
The authors declare no competing interests.
