## [Peer Review File · Communications Biology]

Reviewers' comments:

Reviewer #1 (Remarks to the Author):

- Key results:

Boothe et al extends the fluorescence content-aware image restoration (CARE) method to luminescence microscopy to permit shorter exposure times to improve temporal resolution and throughput. The biggest challenge in bioluminescence microscopy is signal intensity and the most recent advances in luminescence microscopy have addressed this with the development of better probes and the development of better instrumentation. Boothe et al has taken the novel approach to improve this important tool for quantifying an optical reporter by applying the deep neural network, CARE, to restore luminescence recordings without compromising image quality and permitting significant reduction in image exposure time.

- Validity:

The work presented a technique worthy of broad adoption and incorporation into microscopy imaging systems and software. Bioluminescence imaging is a powerful technique well suited for complex tissues with high autofluorescence like the cells of the kidney. Since the claims presented here were straight forward, and the bioluminescent CARE method an extension of a previously published system, the authors do not need to present much information to strengthen their conclusions. It would, however, benefit the reader who does not know the literature supporting the CARE method to do a full description of how the method works to a wider audience and potentially schematics that support these descriptions.

- Originality and significance:

This work is both original and significant, but the application of the method to a biologically important research subject reduces the significance and makes this paper more fit for a specialized journal, not a Nature Portfolio journal. In order to increase the interest in this method, the authors should consider applying this powerful technique to a system that would benefit from the improvements the authors present.

- Data & methodology:

The authors present sufficient data and information on methodology, especially for those skilled in the art. For viewers not skilled in the art, I believe that applying this technique in their hands would not be feasible without more a descriptive method either as a supplemental or a link out. Thus, this work would not be reproducible to the broader audience at the level of detail provided in the methods and in the links provided.

- Appropriate use of statistics and treatment of uncertainties:

It does not appear that any statistical analysis was done to support the authors claims. Both figures 1 and 2 would benefit from statistical analysis.

- Conclusions:

The authors conclusions are satisfactory for the work they presented, however this article would benefit by applying it to a novel research problem to reveal critical data inaccessible with the current state of the art.

- Suggested improvements:

It would greatly benefit this paper by pushing the application of bioluminescence microscopy using the CARE method further by focusing on a disease model where luminescence microscopy would provide data not currently accessible with standard microscopy. For example, Weigert et al used the image from a whole *Drosophila* wing which was a convincing application of the new technology. Repeating this would be important display of the power of this technique. Additionally, imaging tissue sections where single cells need to be quantified and localized such as T and stem cell delivery and infectious disease models would be of broad appeal and application.

- References:

The references cited are satisfactory.

- Clarity and context:

The authors do a satisfactory job in presenting the technology in the abstract and conclusion. However, the introduction needs to appeal to a broader audience by explicitly describing the technology and a figure supporting it.

- Inflammatory material:

No inflammatory material noted.

Reviewer #2 (Remarks to the Author):

In the paper by Boothe et al, they investigated the restoration of bioluminescence images of NanoLuc luciferase expressed in cells by the CARE network. They demonstrated that they were able to restore images of cells with high SNR from raw bioluminescence images taken at short exposure times by the CARE network with various types of training approaches: pairs of short and long exposure images, ones of binned short exposure and unbinned long exposure images, and ones of short exposure images with the noise2noise approach. They also used training with pairs of short exposure bioluminescence and fluorescence images to demonstrate restoration of nuclei and mitochondria images. The results are well founded and the conclusions soundly based on the actual results. As the authors claimed, bioluminescence microscopy using luciferase tag is often advantageous over fluorescence imaging because of no autofluorescence and little influence on optogenetic tools. Although one of its big problems has been the low luminescence intensity, the present study successfully circumvented the problem and achieved the restoration of high-quality bioluminescence images of cells. The manuscript is well written overall, but they should improve it for the convenience of readers (see below). Thus, I recommend the authors to revise the manuscript.

Introduction

In Introduction, they explained the advantages of bioluminescence microscopy over fluorescence microscopy and its drawback of low signal intensities. In addition, the background information of machine learning-based denoising in microscopy had better be summarized briefly so that the readers evaluate what is new in this paper.

Training data

For training the CARE network, they consistently chose the same types of samples as the samples to be observed, and I guess that this would be one limitation of the present technique. I think that they should briefly explain in the text what type of samples should be used for training.

Evaluation of the image restoration through MAE

The authors used the mean absolute error (MAE) to evaluate how the image restoration was

successful. Although MAE seems useful to evaluate the fidelity of restored images, readers are likely to be interested in how fine structures or objects were restored through the CARE network. Thus, I wonder if they could evaluate the spatial resolution of restored images.

Masks in the figures

Mask panels are included in Figures 1-2 and Supplementary Figures 1-2, but they are little explained in the text. The authors should explain what the masks mean to help readers understand the results.

L 59

The author should provide the definition of "technical noise" somewhere in the text.

L 98

The text says, "luminescence microscopy has been predominantly applied to whole tissue or organism imaging since the signals are often too weak for the live imaging of subcellular dynamics". In addition, they actually noticed "the continuous remodeling of the mitochondrial network". Because the authors set the problem in that manner, they should demonstrate "live imaging of subcellular dynamics", and I believe that they are able to present things like time-lapse observation of mitochondria remodeling.

L 116 and Figure 2b right hand

The text says, "we show that restorations of Rotenone treated cells can be distinguished from their DMSO treated controls". Although I can see that in the scattered plots the data points distribute differently between the plot with DMSO and the one with Rotenone, they should also explicitly explain how the plots are read to reach a conclusion of "distinguished" to make sure readers' understanding, not just saying "distinguished".

Reviewer #1 (Remarks to the Author):

- Key results:

Boothe et al extends the fluorescence content-aware image restoration (CARE) method to luminescence microscopy to permit shorter exposure times to improve temporal resolution and throughput. The biggest challenge in bioluminescence microscopy is signal intensity and the most recent advances in luminescence microscopy have addressed this with the development of better probes and the development of better instrumentation. Boothe et al has taken the novel approach to improve this important tool for quantifying an optical reporter by applying the deep neural network, CARE, to restore luminescence recordings without compromising image quality and permitting significant reduction in image exposure time.

- Validity:

The work presented a technique worthy of broad adoption and incorporation into microscopy imaging systems and software. Bioluminescence imaging is a powerful technique well suited for complex tissues with high autofluorescence like the cells of the kidney. Since the claims presented here were straight forward, and the bioluminescent CARE method an extension of a previously published system, the authors do not need to present much information to strengthen their conclusions.

Thank you.

1. It would, however, benefit the reader who does not know the literature supporting the CARE method to do a full description of how the method works to a wider audience and potentially schematics that support these descriptions.

Done. We have added a schematic to the supplementary figures (S3) to illustrate the workflow:

1. record training data pairs
2. train CARE network
4. restore experimental data

Further, we provided more detail on the procedures in the methods section of the manuscript. The text now reads (line 355 ff):

For training and restoration, the previously published CSBDeep package v0.6.0 was used (<http://csbdeep.bioimagecomputing.com>). A detailed documentation of this software is available at <http://csbdeep.bioimagecomputing.com/doc>. In general, the presented method consists of 4 steps (Figure S3): First, training data pairs need to be recorded (Figure 1 a-c). They should be as similar as possible to the experimental data that will be restored later. Second, the CARE network needs to be trained. Third, the experimental data can be recorded, which consists only of noisy image data. Fourth, the experimental data can be restored using the previously trained CARE network.

- Originality and significance:

2. This work is both original and significant, but the application of the method to a biologically important research subject reduces the significance and makes this paper more fit for a specialized journal, not a Nature Portfolio journal. In order to increase the interest in this method, the authors should consider applying this powerful technique to a system that would benefit from the improvements the authors present.

Done. First, we have applied our method to whole organism luminescence imaging in mice (Figure 1d, line 94 ff) and demonstrate an order of magnitude reduction in necessary exposure times from 20 s to 2 s.

The practical significance of this result is that it reduces luminescence exposure times within the range where luminescence imaging of free-ranging rodents becomes conceptually feasible. We discuss this possibility in the revised discussion, which now reads (line 189 ff):

In the case of in vivo luminescence imaging, the exposure time reductions are similarly significant. The up to 10-fold exposure time reduction that we achieved on the basis of a standard firefly luciferase reporter and a conventional imaging system (Figure 1d) can certainly be reduced further via the use of enhanced luciferase reporters and optimised imaging setups.¹⁵ Hence we suggest, that exposure times in the sub-second range are feasible, which in turn might enable the luminescence imaging of free-ranging rodent models without the need for anaesthetics.

Second, we have added new time course luminescence imaging data to demonstrate the utility of our method in the reconstruction of subcellular dynamics (Figure 2 c):

To the best of our knowledge, this is the first time that mitochondria dynamics can be visualized using conventional luminescence microscopy. Our method is therefore enabling the excitation light free study of mitochondria dynamics in especially photosensitive tissue cultures. The revised version of the manuscript conclusively demonstrates the real-world significance of our method in a broad range of experimental systems.

- Data & methodology:

3. The authors present sufficient data and information on methodology, especially for those skilled in the art. For viewers not skilled in the art, I believe that applying this technique in their hands would not be feasible without more a descriptive method either as a supplemental or a link out. Thus, this work would not be reproducible to the broader audience at the level of detail provided in the methods and in the links provided.

Done. As explained before (#1), we now provide more detail on the methodology. Further, we also provide in an external repository (https://gitlab.gwdg.de/rinklab_public/lumicare) a fully functional test environment including sample data and scripts, which are annotated and explained in detail. We believe that this provides a comprehensive resource for users that are new to the topic and makes it easy to apply the method to their own experimental questions. We refer to this repository in line 364 of the manuscript, which reads:

Generalised Python scripts that can be used with this container for preparing training data, network training and prediction are available at https://gitlab.gwdg.de/rinklab_public/lumicare.

- Appropriate use of statistics and treatment of uncertainties:

4. It does not appear that any statistical analysis was done to support the authors claims. Both figures 1 and 2 would benefit from statistical analysis.

Done. We thank the reviewer for this critical and very valuable comment.

In Figure 1 we statistically evaluate the data by directly reporting mean absolute errors. We do not believe that it is meaningful to perform statistical tests on statistical values. Further, it needs to be considered, that the users need to decide for themselves how much error is tolerable. E.g. what might be insignificantly different on paper is maybe still not a good enough restoration for the user. The opposite is equally plausible when a user only needs a restoration of instances (e.g. number of cells) and is not dependent on a very precise restoration of morphological details. Which restoration accuracy is needed for the respective application can differ quite widely and is not necessarily tied to statistical significance of the restoration accuracy. We therefore do not apply statistical tests for this particular analysis performed in Figure 1.

Since we report a direct readout (Pearson Correlation) in Figure 2a it is correct that a statistical analysis could be applied. In the legend for Figure 2a (line 223) we now provide the p-value proving no statistically significant difference between restorations with fluorescence or restorations with long exposure luminescence as ground truth. The previous decision boundary in Figure 2b was a manual fit delineating one condition entirely and therefore had no statistical foundation. We now use in a revised version of Figure 2b the statistical method of linear regression to fit a decision boundary to separate the experimental conditions:

• Conclusions:

- The authors conclusions are satisfactory for the work they presented, however this article would benefit by applying it to a novel research problem to reveal critical data inaccessible with the current state of the art.

Done. Please see response #2 above.

• Suggested improvements:

- It would greatly benefit this paper by pushing the application of bioluminescence microscopy using the CARE method further by focusing on a disease model where luminescence microscopy would provide data not currently accessible with standard microscopy. For example, Weigert et al used the image from a whole *Drosophila* wing which was a convincing application of the new technology. Repeating this would be important display of the power of this technique. Additionally, imaging tissue sections where single cells need to be quantified and localized such as T and stem cell delivery and infectious disease models would be of broad appeal and application.

Thank you for this suggestion. We think that the added *in vivo* luminescence reporter imaging in mice (see above) already partially addresses this point, given the many tumor biology or infectious disease applications in the system that can benefit from shorter luminescence exposures. Our own model system, planarian flatworms, are still barely amenable to transgene expression. Unfortunately, we do not have access to suitable *Drosophila* lines, fly culture infrastructure and required skills to perform the suggested experiment.

We hope to provide with our new data convincing evidence that our method is widely applicable across biological scales and model systems.

- References:

The references cited are satisfactory.

- Clarity and context:

7. The authors do a satisfactory job in presenting the technology in the abstract and conclusion. However, the introduction needs to appeal to a broader audience by explicitly describing the technology and a figure supporting it.

Answer: We thank the reviewer for the feedback and added a brief explanation of the technologies working principle to the introduction (line 52 ff), which now reads:

In this method, we train a deep neural network with image pairs consisting of low and high signal to noise recordings. From this, the network learns to denoise images of low signal quality and ultimately enhances the image contrast in a content aware manner.

In addition, we added a graphical visualization of the method and workflow as supplemental figure S3 for and we expanded the respective part of the methods section (line 358 ff). This now reads:

In general, the presented method consists of 4 steps (Figure S3): First, training data pairs need to be recorded (Figure 1 a-c). They should be as similar as possible to the experimental data that will be restored later. Second, the CARE network needs to be trained. Third, the experimental data can be recorded, which consists only of noisy image data. Fourth, the experimental data can be restored using the previously trained CARE network.

- Inflammatory material:

No inflammatory material noted.

Reviewer #2 (Remarks to the Author):

In the paper by Boothe et al, they investigated the restoration of bioluminescence images of NanoLuc luciferase expressed in cells by the CARE network. They demonstrated that they were able to restore images of cells with high SNR from raw bioluminescence images taken at short exposure times by the CARE network with various types of training approaches: pairs of short and long exposure images, ones of binned short exposure and unbinned long exposure images, and ones of short exposure images with the noise2noise approach. They also used training with pairs of short exposure bioluminescence and fluorescence images to demonstrate restoration of nuclei and mitochondria images. The results are well founded and the conclusions soundly based on the actual results. As the authors claimed, bioluminescence microscopy using luciferase tag is often advantageous over fluorescence imaging because of no autofluorescence and little influence on optogenetic tools. Although one of its big problems has been the low luminescence intensity, the present study successfully circumvented the problem and achieved the restoration of high-quality bioluminescence images of cells. The manuscript is well written overall, but they should improve it for the convenience of readers (see below). Thus, I recommend the authors to revise the manuscript.

- Introduction

1. In Introduction, they explained the advantages of bioluminescence microscopy over fluorescence microscopy and its drawback of low signal intensities. In addition, the background information of machine learning-based denoising in microscopy had better be summarized briefly so that the readers evaluate what is new in this paper.

Answer: We thank the reviewer for the feedback and added a brief explanation of the technologies working principle to the introduction (line 52 ff), which now reads:

In this method, we train a deep neural network with image pairs consisting of low and high signal to noise recordings. From this, the network learns to denoise images of low signal quality and ultimately enhances the image contrast in a content aware manner.

In addition, we added a graphical visualization of the method and workflow as supplemental figure S3 for and we expanded the respective part of the methods section (line 358 ff). This now reads:

In general, the presented method consists of 4 steps (Figure S3): First, training data pairs need to be recorded (Figure 1 a-c). They should be as similar as possible to the experimental data that will be restored later. Second, the CARE network needs to be trained. Third, the experimental data can be recorded, which consists only of noisy image data. Fourth, the experimental data can be restored using the previously trained CARE network.

1. record training data pairs
2. train CARE network
3. record experimental data
4. restore experimental data

- Training data

2. For training the CARE network, they consistently chose the same types of samples as the samples to be observed, and I guess that this would be one limitation of the present technique. I think that they should briefly explain in the text what type of samples should be used for training.

Done and thanks for pointing out this seeming caveat. Our method is in principle applicable to any sample type of any specimen. To support this statement we now added data from mice as an example for an *in vivo* system (Figure 1d):

This example also highlights that our method can be applied equally well to 2D data. Further, we demonstrate that not only still images but also time-lapse recordings can be restored with high accuracy (Figure 2 c):

Together we hope that these additional data convince the reviewer that our approach is not limited to a single sample type.

- Evaluation of the image restoration through MAE

3. The authors used the mean absolute error (MAE) to evaluate how the image restoration was successful. Although MAE seems useful to evaluate the fidelity of restored images, readers are likely to be interested in how fine structures or objects were restored through the CARE network. Thus, I wonder if they could evaluate the spatial resolution of restored images. **Answer: CARE restores images on a per-pixel basis (Weigert et. al 2018). Therefore, the resolution of the restored images is entirely dependent on the resolution of the training data. This has been extensively discussed in the initial publication of CARE in which Weigert et al show, that even structures below the resolution limit of diffraction limited standard optical systems can in principle be restored. Since we highlight in the text clearly that we use CARE, we believe that it is not necessary to evaluate the resolution capabilities of this technique in our manuscript. Nevertheless, in the newly added figure 2c it is evident that fine mitochondrial structures, which are very few pixels wide, can be reliably restored. It highlights that the resolution of the restored image is ultimately dependent on the resolution of the training data. An assessment of the resolution capabilities of CARE is therefore not required. To support this, we would like to introduce the reviewer to a data set that is not part of this work but nicely demonstrates CARE's per-pixel restoration capabilities. Below we trained a network with conventional widefield images paired with their deconvolved counterpart. Deconvolution is a plain mathematical procedure transforming each input pixel value into an output value based on a globally valid algorithm. CARE learns this arithmetic transformation with extremely high precision creating restorations that are down to a single pixel virtually indistinguishable from a conventionally deconvolved input image:**

- Masks in the figures

4. Mask panels are included in Figures 1-2 and Supplementary Figures 1-2, but they are little explained in the text. The authors should explain what the masks mean to help readers understand the results.

Done. We agree and added additional information accordingly to explain more clearly the generation and meaning of the masks (line 71 ff). The respective text now reads:
To quantify the quality of the restorations, all images were automatically thresholded resulting in binary masks representing the objects of interest. The disagreement of masks obtained from input, restoration and ground truth images was quantified with the mean absolute error (MAE).

5. L 59 - The author should provide the definition of “technical noise” somewhere in the text
Done. The corresponding section in the main text has been rephrased and now reads (line 74 ff):

The restorations from the aforementioned recordings display a MAE that is similar to the noise between object masks from two subsequently acquired images at 60 s exposure time (“technical noise”).

6. L 98 - The text says, “luminescence microscopy has been predominantly applied to whole tissue or organism imaging since the signals are often too weak for the live imaging of subcellular dynamics”. In addition, they actually noticed “the continuous remodeling of the mitochondrial network”. Because the authors set the problem in that manner, they should demonstrate “live imaging of subcellular dynamics”, and I believe that they are able to present things like time-lapse observation of mitochondria remodeling.

Done. We agree with the reviewer and now present a new panel that shows the successful restoration of mitochondria dynamics in time-lapse recordings (Figure 2c):

7. L 116 and Figure 2b right hand - The text says, “we show that restorations of Rotenone treated cells can be distinguished from their DMSO treated controls”. Although I can see that in the scattered plots the data points distribute differently between the plot with DMSO and the one with Rotenone, they should also explicitly explain how the plots are read to reach a conclusion of “distinguished” to make sure readers’ understanding, not just saying “distinguished”.

Done and thanks for pointing out this flaw. To make the distinction between the treatments statistically valid, we now updated the graphs by fitting a decision boundary via linear regression into the datasets (Figure 2b):

The fitted decision boundary separates both experimental groups (DMSO vs Rotenone) similarly well when comparing ground truth fluorescence data with restored luminescence data. The text and figure legends were adjusted accordingly.

REVIEWERS' COMMENTS:

Reviewer #2 (Remarks to the Author):

I applaud Booth et al in their work and revision of this manuscript. All of my concerns were met in this revision. These findings and shared techniques will be of broad appeal and high impact in the field of in vivo imaging. I strongly recommend for immediate publication.

Reviewer #3 (Remarks to the Author):

The revised manuscript by Boothe et al. has been greatly improved. They addressed my concern about the manuscript, although I have a very small comment: in the graphs in Figure 2b, the data point for Rotenone in the legend should be enclosed by a dark circle like the data points in the graphs for consistency and visibility. Except for this, I do not have any comments about the manuscript. After revising this, I recommend this paper for publication.